# Turkish adaptation of a new scale for measuring transactional distance between students and the learning technology

Alperen Yandi[ID]*

Faculty of Education, Bolu Abant Izzet Baysal University, Golkoy Campus, Bolu, Turkey

* alperenyandi@ibu.edu.tr

## Abstract

Transactional distance is a key construct for understanding the quality of interactions in distance education. This study aimed to adapt the New Scale for Measuring Transactional Distance Between Students and Technology to Turkish culture. The adaptation process followed the 18-step guideline by the International Test Commission (ITC). The scale was applied to two study groups: Study Group 1 (n = 46) for linguistic equivalence and preliminary analysis, and Study Group 2 (n = 2402) for validation. Confirmatory factor analysis confirmed the four-factor structure with acceptable fit indices (CFI = 0.964, TLI = 0.947, RMSEA = 0.079, SRMR = 0.049). The reliability coefficients (Cronbach's α) ranged from 0.702 to 0.939 across dimensions. Discriminant validity was supported through HTMT2 values, all below 0.90. The results suggest that the adapted scale is a valid and reliable tool for measuring transactional distance in Turkish distance education contexts.

## Introduction

Distance learning has a long history, with examples dating back to the 20th century [1]. Its recent growth accelerated during the SARS-CoV-2 pandemic, which led to widespread adoption of remote or hybrid instruction. However, fully planned distance education differs from "emergency remote teaching" implemented as a crisis response [2]. The shift brought attention to the importance of instructional design, which was often overlooked during emergency transitions.

The rapid move to online environments prompted extensive research and innovations in educational practice [3]. Data from the National Center for Education Statistics and the National Council for State Authorization Reciprocity Agreements show a clear rise in online enrollment. Systematic reviews also emphasize the role of educational technology and instructional strategies in supporting this transformation [4].

Many studies have explored factors affecting the quality of distance learning and student attrition [5–8]. Among these, technological proficiency plays a central role by influencing student–teacher and student–student interactions [9].

**Data availability statement:** All relevant data are uploaded to Zenodo (https://doi.org/10.5281/zenodo.16880975).

**Funding:** The author(s) received no specific funding for this work.

**Competing interests:** The authors have declared that no competing interests exist.

Transactional distance theory provides a theoretical basis for these interactions. It defines transactional distance as the pedagogical gap between instructor and learner, shaped by three variables: structure, dialogue, and autonomy [10]. In distance education, minimizing this gap depends on effective dialogue and thoughtful course design.

Structure refers to the flexibility of course elements like content and assessment, while dialogue encompasses open communication between participants [11]. Autonomy relates to the learner's control over the learning process. These elements interact to define the flexibility and complexity of instruction [10,12]

Viewed psychometrically, transactional distance has been measured using self-report tools [13–16]. High transactional distance can negatively affect learning [17], prompting researchers to examine the construct more closely. Sub-dimensions such as student–student, student–teacher, and student–content have been proposed [10]. Greater interaction in these areas corresponds to lower transactional distance and supports meaningful learning [18].

Despite its importance, the student–technology interface was long overlooked [9]. Eventually, this interaction became a fourth sub-dimension, focusing on users' understanding and control of digital interfaces. Early efforts to measure this included tools by Zhang [17], but these became outdated as technology evolved [19]. Paul [15] observed that such tools often yielded overly positive responses.

This led to the development of new instruments. Weidlich and Bastiaens [16] proposed a scale measuring perceived transactional distance in terms of students' basic technological competence and their perceptions of usability. The former draws from Hung et al.'s [20] "Online Learning Readiness Scale," specifically its "Computer and Internet Self-Efficacy" items. The latter aligns with ISO's definition of usability [21].

To ensure relevance across diverse cultures, such scales must be adapted appropriately. In Turkey, Zhang's [17] scale was adapted by Yılmaz and Keser [22], and Horzum [23] developed a transactional distance scale with five sub-factors. However, these tools predate the recent acceleration in online learning and omit the student–interface dimension.

The aim of this study is to adapt the new scale developed by Weidlich and Bastiaens [16], which includes student-interface interaction, to Turkish culture and evaluate its psychometric properties.

This adaptation is preferred over developing a new instrument for three reasons: the scale already reflects key contemporary constructs; technology use often transcends cultural boundaries; and adapting a validated tool is more efficient and cost-effective than building one from scratch [24].

Cross-cultural equivalence enhances the accuracy of international comparisons and supports broader research collaboration. This study therefore asks

What is the reliability and validity of the four-factor structure of the new transactional distance scale in Turkish culture?

## Method

This section contains information about the steps of adaptation process. In this way, it aims to provide other researchers with a detailed of the process. It also includes

information regarding the characteristics of the sample and the adapted measurement instrument. Furthermore, it encompasses all analysis procedures conducted during the adaptation steps. The data collection process for the study was carried out between November 29, 2021, and December 25, 2021.

## The adaptation process

The adaptation was guided by the 18-step procedure recommended by the International Test Commission [25]. These steps span six categories: Pre-Condition (PC), Test Development (TD), Confirmation (C), Administration (A), Scoring and Interpretation (SSI), and Documentation (Doc). Each step was addressed with the exception of Steps 12 and 16, which require multilingual samples. These were not feasible due to the lack of access to large-scale cross-cultural data and limitations in the original developers' dataset. Follow-up studies may address these gaps.

PC-1 to PC-3: Permissions were obtained from the original authors. Conceptual overlap and cultural equivalence were confirmed through expert evaluations and item-by-item review.

TD-1 to TD-5: The translation process used a forward–backward method with contributions from 13 linguists. Items were adapted to reflect local usage (e.g., "Moodle" was replaced with widely used platforms such as "LMS, Google Classroom, MS Teams"). Pilot data were collected to perform item analysis.

C-1 to C-4: Sampling was based on representativeness and adequate size. Statistical evidence was gathered to support construct equivalence and reliability. Equating procedures (Step 12) were not applied due to lack of comparable language versions.

A-1 to A-2: The administration was performed online with instructions tailored to minimize cultural misunderstanding. Participants could skip items they chose not to answer.

SSI-1 to SSI-2: Equivalence in responses between source and target language versions was assessed via correlation and t-tests. Cross-population comparison (Step 16) was not feasible.

Doc-1 to Doc-2: A technical manual and practical guidelines for users were developed and included in the appendices.

Further details corresponding to each ITC step and supporting analyses are presented in the Results section.

## Study groups

For the adaptation process, data were collected from two different study groups. The first was Study Group 1, in which the pilot application was conducted to ensure linguistic equivalence and identify possible arrangements; the other is Study Group 2, which was formed to obtain empirical evidence for the adapted scale.

**Study group 1.** Study Group 1 consisted of 46 fourth-year students enrolled in a foreign language teaching department in Turkey. This group was selected for their advanced language proficiency in both English and Turkish, allowing for reliable assessment of linguistic equivalence. Participants' ages ranged from 21 to 31 years (M = 22.76, SD = 1.97). The majority were female (78.30%). Detailed demographic data are provided in Table 1.

**The study group 2.** Study Group 2 comprised 2402 university students enrolled in various faculties. This group was used to conduct large-scale statistical validation. After outlier removal procedures, the final sample included 2269 participants. The average age was 22.47 years (SD = 4.20), with 66.90% identifying as female. Additionally, 82.60% reported owning a personal computer, and 87.50% were enrolled in undergraduate (Bachelor's degree) programs. Full demographic characteristics are summarized in Table 2.

Determination of an adequate sample size is critical for ensuring the validity and reliability of psychometric analyses, particularly in confirmatory factor analysis (CFA) and measurement invariance testing. According to MacCallum et al. [26], sample sizes of approximately 200 are considered adequate for CFA when communalities are high and model fit is good. Furthermore, Wolf et al. [27] recommend that for models with moderate complexity and latent variables, a minimum sample size of 200–500 participants is generally sufficient to achieve stable parameter estimates and adequate statistical power.

**Table 1. Demographic information of study group 1.**

| Gender | Frequencies | Percentage |
|---|---|---|
| Female | 36 | 78.30 |
| Male | 10 | 21.70 |
| Total | 46 | 100.00 |
| **Age** | **Frequencies** | **Percentage** |
| 21 | 3 | 6.52 |
| 22 | 27 | 58.70 |
| 23 | 10 | 21.74 |
| 24 | 2 | 4.35 |
| 25 | 2 | 4.35 |
| 31 | 2 | 4.35 |
| Total | 46 | 100.00 |
| Mean | 22.760 | |
| Standard Deviation | 1.968 | |

In the context of structural equation modeling (SEM), Kline [28] suggests a rule of thumb of at least 10 observations per estimated parameter, or an absolute minimum of 200 observations, whichever is larger. Given that the final sample size for Study Group 2 was 2269 participants, it substantially exceeds these recommended thresholds, thereby providing robust power for CFA and multi-group analyses. This large sample size supports the precision of parameter estimates and increases confidence in the generalizability of the findings within the target population.

## Measurement instrument

The scale adapted in this study is the "New TDSTECH," originally developed by Weidlich and Bastiaens [16]. It includes 11 items designed to measure transactional distance between students and the technology used in online learning environments. The scale is based on two core dimensions: (1) the learner's basic technological competence, and (2) the perceived design and usability of the learning platform.

The first dimension draws on the "Computer and Internet Self-Efficacy" subscale of the Online Learning Readiness Scale developed by Hung et al. [20]. The second dimension reflects the International Organization for Standardization's (ISO) definition of usability: the extent to which a product can be used effectively, efficiently, and satisfactorily by specified users.

During translation, references to "Moodle" were replaced with the broader phrase "online learning platforms (e.g., LMS, Moodle, MS Teams, Google Classroom)" to reflect the diversity of systems used in the Turkish context. This ensured cultural and contextual alignment without altering item intent.

Following factor analysis conducted by the original developers, the final version includes four components: learner readiness to use technology, perceived effectiveness, efficiency, and satisfaction. Each of the first, second, and fourth components consists of three items, while the efficiency component includes two items (Items 7 and 8), both of which are negatively worded.

The items use a five-point Likert scale ranging from "strongly disagree" to "strongly agree." Internal consistency reported by the original authors was high ($\alpha = 0.873$). The item distribution across components is summarized in Table 3.

## Analysis of the data

All statistical analyses were conducted using SPSS 20.0, Mplus 7.3, and R 4.2.1 [29], along with several R packages: lavaan [30] for structural equation modeling, semTools [31] for measurement invariance testing, cSEM [32] for HTMT2

**Table 2. Demographic information of study group 2.**

| Gender | Frequencies | Percentage |
|---|---|---|
| **Female** | 1608 | 66.90 |
| **Male** | 794 | 33.10 |
| **Total** | 2402 | 100.00 |
| **Personal Computer** | **Frequencies** | **Percentage** |
| **Having** | 1983 | 82.60 |
| **Not Having** | 419 | 17.40 |
| **Total** | 2402 | 100.00 |
| **Study Period** | **Frequencies** | **Percentage** |
| **Associate degree** | 300 | 12.50 |
| **Bachelor's degree students** | 2102 | 87.50 |
| **Total** | 2402 | 100.00 |
| **Age** | **Frequencies** | **Percentage** |
| 17 | 2 | 0.10 |
| 18 | 67 | 2.80 |
| 19 | 198 | 8.20 |
| 20 | 466 | 19.40 |
| 21 | 507 | 21.10 |
| 22 | 461 | 19.20 |
| 23 | 240 | 10.00 |
| 24 | 129 | 5.40 |
| 25 | 74 | 3.10 |
| 26 | 33 | 1.40 |
| 27 | 28 | 1.20 |
| 28 | 29 | 1.20 |
| 29 | 19 | 0.80 |
| 30 | 9 | 0.40 |
| 31 | 13 | 0.50 |
| 32 | 11 | 0.50 |
| 33 | 7 | 0.30 |
| 34 | 13 | 0.50 |
| 35 | 14 | 0.60 |
| 36 | 11 | 0.50 |
| 37 | 13 | 0.50 |
| 38 | 9 | 0.40 |
| 39 | 11 | 0.50 |
| 40 | 16 | 0.70 |
| 41 | 9 | 0.40 |
| 42 | 6 | 0.20 |
| 45 | 7 | 0.30 |
| **Total** | 2402 | 100.00 |
| **Mean** | 22.470 | |
| **Standard Deviation** | 4.197 | |

**Table 3. Distribution of the items for the four-component construct of the new TDSTECH.**

| Item code | Four-Component Construct |
|-----------|--------------------------|
| y1 | Component 1 (C1) |
| y2 | |
| y3 | |
| y4 | Component 2 (C2) |
| y5 | |
| y6 | |
| y7 | Component 3 (C3) |
| y8 | |
| y9 | Component 4 (C4) |
| y10 | |
| y11 | |

*Note. C1 = Learner readiness to use technology; C2 = Effectiveness; C3 = Efficiency (negatively worded items); C4 = Satisfaction.

calculation, and dplyr [33] and corrplot [34] for data wrangling and visualization. The significance level for all tests was set at.05.

**Assumption checks and data screening.** Prior to assumption testing, the dataset was checked for missing values, inconsistent entries, and reverse-coded items. Items 7 and 8, which were negatively worded, were reverse scored before any analyses. Participants with missing responses exceeding 10% of the items were excluded. Data cleaning was conducted using listwise deletion, and descriptive screening confirmed that all retained variables were within acceptable limits for missingness.

Before confirmatory analyses, the dataset from Study Group 2 was examined for univariate and multivariate outliers. Univariate outliers were identified using z-standardized values for each item. Participants with scores beyond ±3.29 were removed [35]. Subsequently, Mahalanobis distance was calculated to identify multivariate outliers [36]. Participants with values exceeding the chi-square critical value at p < .001 were excluded. As a result, 133 responses were removed, and analyses continued with 2269.

**Tests of normality and linearity.** Univariate normality was assessed using standardized skewness and kurtosis values. Values within ±1.96 were considered indicative of normality at the 5% significance level [35,37]. Some items exceeded this threshold, suggesting deviations from univariate normality.

Multivariate normality was tested using Mardia's coefficient [36] and the results indicated significant multivariate non-normality.

To examine linearity, scatter plots of item correlations were reviewed. Bartlett's Test of Sphericity ($\chi^2(55) = 18275.057$, p < .001) indicated that correlations were sufficiently large for factor analysis [38].

**Multicollinearity assessment.** Multicollinearity and singularity were assessed through Variance Inflation Factor (VIF) and tolerance values. VIF values ranged from 1.380 to 5.200, and tolerance values ranged from.1902 to.720, indicating no multicollinearity concerns [28,38].

**Justification for estimator choice.** Due to observed multivariate non-normality, the robust maximum likelihood estimation method (MLR) was selected for CFA and MGCFA procedures. MLR adjusts standard errors and fit indices to account for non-normality and is recommended under such conditions [39–41].

**Residual assumptions.** To ensure the appropriateness of residuals under MLR estimation, the following checks were conducted:

- Q–Q plot of standardized residuals showed approximate normality in the center but minor deviations in the tails (Fig 1) [42].

   The figure displays the quantile–quantile (Q–Q) plot of standardized residuals. The points closely follow the diagonal line, indicating approximate normality in the center with minor deviations in the tails.

- Linearity between predicted and residual values was visually confirmed via scatter plot (Fig 2) [28].

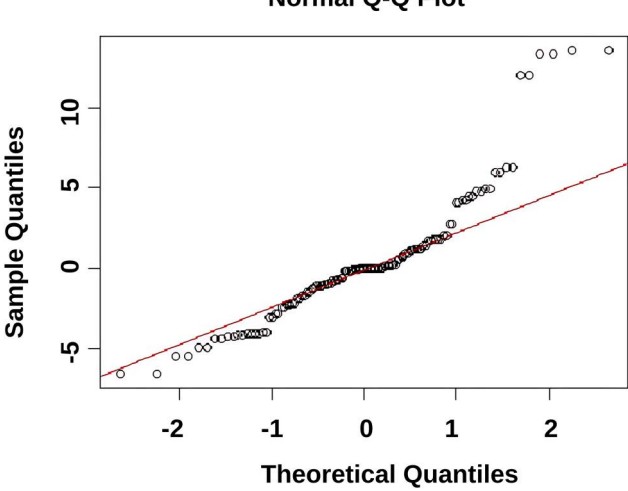

**Fig 1. Q–Q plot of standardized residuals from the confirmatory factor analysis model.**

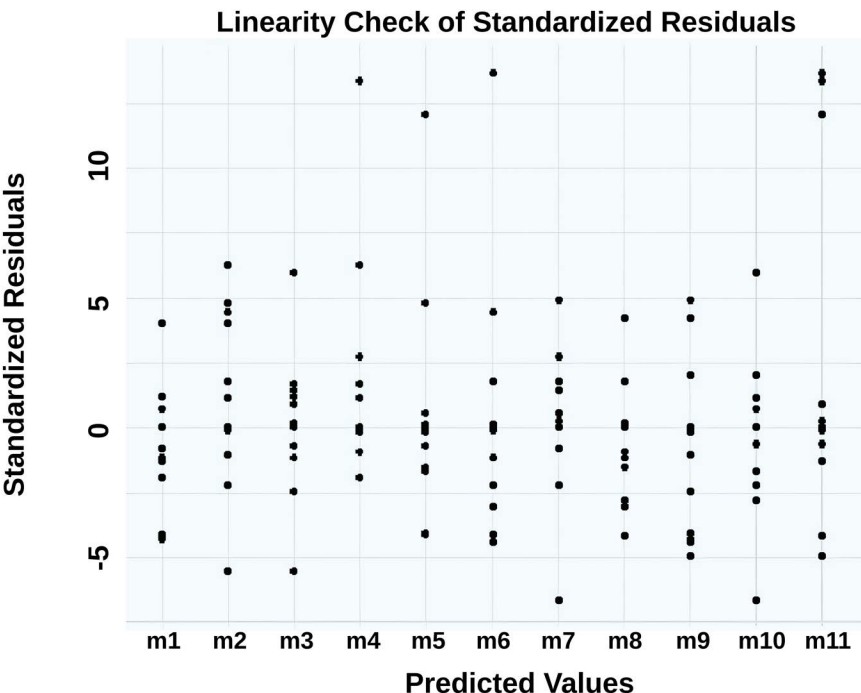

**Fig 2. Scatter plot of standardized residuals from the confirmatory factor analysis model.**

**The scatter plot illustrates the relationship between predicted values and standardized residuals. The random distribution of points suggests linearity and absence of systematic bias.**

- Homoscedasticity was tested using the Breusch–Pagan test (BP = 3.6517, df = 1, p = .05601), indicating non-significant heteroscedasticity [43].

- Autocorrelation was assessed via Durbin–Watson test (DW = 1.7604, p = .09192), showing no significant autocorrelation [35,44,45].

### Analysis flow

After confirming estimation assumptions, CFA was used to assess the structural validity of the four-factor model. Model fit was evaluated using the following indices and cut-off values: $\chi^2$/df (< 2 for good fit), RMSEA (<.08), CFI and TLI (≥.95), and SRMR (<.06) [46–48].

Measurement invariance across gender, study program, and computer ownership was tested via MGCFA using the stepwise approach [49–51]. Configural, metric, and scalar models were compared using ΔCFI (±.01), ΔTLI (±.01), ΔRMSEA (<.015), and ΔSRMR (<.030) thresholds [28].

Internal consistency was evaluated using Cronbach's alpha, and discriminant validity was assessed via HTMT2 (threshold < .90) [52–54].

All results are reported in the Results section

### Ethics statement

This study received approval from the Ethics Committee of Bolu Abant Izzet Baysal University (Approval Number: NO. 2021/430 [29.11.2021 and the meeting number: 2021/11]), and all procedures adhered to the ethical standards set by institutional and national research committees, along with the 1964 Helsinki Declaration and its subsequent amendments. Before participation, all participants received a written online information sheet outlining the study's goal, the voluntary nature of participation, data confidentiality, and the possibility to withdraw at any time without penalty. Participants electronically supplied written informed permission by actively signalling their acceptance using the online survey system prior to completing the questionnaire. Only the data from participants who granted this consent were incorporated into the study. This research did not include minors; hence, parental or guardian consent was unnecessary.

### Results

This section presents the findings obtained in each step of the adjustment process. The results of the statistical analysis were interpreted in accordance with the theoretical findings. Although the presentations for some steps are not results, the information about those steps is provided as a heading to ensure completeness of all steps.

**PC-1 (1) Obtain the necessary permission from the holder of the intellectual property rights relating to the test before carrying out any adaptation.**

One of the original developers of the instrument was contacted via email to obtain formal permission for the adaptation of the scale. The correspondence included a detailed explanation of the purpose and scope of the current study. The author granted written approval for the adaptation process, thereby fulfilling the requirement outlined in PC-1 of the ITC guidelines.

**PC-2 (2) Evaluate that the amount of overlap in the definition and content of the construct measured by the test and the item content in the populations of interest is sufficient for the intended use (or uses) of the scores.**

The concept of transactional distance, as represented in the new TDSTECH model, plays a central role in understanding learners' interaction with technology in distance education contexts. With the rapid expansion of distance learning

programs across cultures, the universality of this construct has become more evident. Similarities in the scale items, application procedures, and learner feedback suggest conceptual alignment between the source and target cultures. Moreover, distance education technologies tend to exhibit standardized structures and functions that transcend cultural boundaries. This high level of overlap supports the assumption that participant scores may be interpreted meaningfully and comparably across culturally similar populations, thereby justifying the scale's use in the target setting.

**PC-3 (3) Minimize the influence of any cultural and linguistic differences that are irrelevant to the intended uses of the test in the populations of interest.**

In this step, the scale items were carefully reviewed to identify and minimize the impact of cultural or linguistic elements that may not be relevant to the intended construct in the target population. A comparative analysis was conducted between the source and target cultures to ensure that the items retained conceptual clarity while remaining culturally appropriate. The only culturally bound reference identified was the specific mention of "Moodle" — an educational technology platform — in multiple items.

Given that Moodle is not the only platform commonly used in the Turkish context, it was replaced with more inclusive terminology to enhance participant familiarity and comprehension. In Item 3, the phrase was modified to "online learning platforms (e.g., LMS, Moodle, MS Teams, Google Classroom)" to provide illustrative examples. In subsequent items where Moodle appeared, only the phrase "online learning platforms" was retained and grammatically adapted to the sentence structure. This selective phrasing was applied to preserve the intended meaning while avoiding redundancy and minimizing potential response fatigue during administration.

**TD-1 (4) Ensure that the translation and adaptation processes consider linguistic, psychological, and cultural differences in the intended populations through the choice of experts with relevant expertise.**

In this step, experts were selected to ensure that the translation and adaptation processes addressed the linguistic, psychological, and cultural dimensions relevant to the target population. A total of 13 linguists, all proficient in both the source and target languages, contributed at various stages of the process. Specifically, two linguists participated in the forward–backward translation phase, eight were consulted during the post-forward translation evaluation, and one linguist with deep cultural fluency conducted the final review. This multidisciplinary involvement aimed to maximize equivalence and clarity in the adapted version.

**TD-2 (5) Use appropriate translation designs and procedures to maximize the suitability of the test adaptation in the intended populations.**

The translation design followed the recommendations of Van De Vijver and Poortinga [55], which emphasize minimizing construct-irrelevant variance during adaptation. Given the conceptual similarity of the construct across cultures, a forward–backward translation method was applied. Linguists were consulted throughout the process to ensure conceptual, semantic, and idiomatic equivalence. Terms were revised to reflect the most culturally and contextually appropriate wording without altering the intended meaning. These adjustments aimed to ensure that participants in the target population would interpret the items in a manner consistent with the source population. To support this equivalence, statistical evidence was gathered through analyses with data from Study Group 1.

**TD-3 (6) Provide evidence that the test instructions and item content have similar meaning for all intended populations.**

As an extension of the translation process, an expert fluent in both languages and familiar with both cultural contexts reviewed the final version. This expert assisted in evaluating the semantic consistency of the item content. In addition to qualitative review, a quantitative comparison was conducted. Specifically, Pearson correlation analysis was performed on the scores from the forward and backward translated forms administered to Study Group 1. The results yielded a coefficient of $r = 0.685$ ($p < .05$), indicating a moderate to strong positive relationship. Although slightly below the commonly accepted benchmark of .70, the result nonetheless suggests a meaningful degree of semantic overlap and supports the assumption that the items hold similar meaning for respondents from both cultures.

**TD-4 (7) Provide evidence that the item formats, rating scales, scoring categories, test conventions, modes of administration, and other procedures are suitable for all intended populations.**

The item formats, response scales, scoring conventions, and administration procedures of the adapted test were retained in a form consistent with the original instrument. No structural changes were made to the Likert-type format or scoring method. Semantic equivalence between the original and adapted items further supports the appropriateness of these procedures for use in the target culture.

**TD-5 (8) Collect pilot data on the adapted test to enable item analysis, reliability assessment, and small-scale validity studies so that any necessary revisions to the adapted test can be made.**

Pilot testing of the adapted scale was conducted using data from Study Group 1 to examine item quality and preliminary psychometric properties. Item-total correlation coefficients ranged from.495 to.830, all exceeding the accepted threshold of.300 [35] indicating adequate item discrimination. Reliability was evaluated using Cronbach's alpha, yielding values of.674,.938,.837, and.848 for the four subdimensions. These results fall within or above the acceptable range of.60–.70 [56] suggesting satisfactory internal consistency in the preliminary sample.

**C-1 (9) Select sample with characteristics that are relevant for the intended use of the test and of sufficient size and relevance for the empirical analyses.**

For the validity and reliability analyses, a sufficiently large and representative sample was selected. The scale was administered to 2402 university students. After the removal of 133 outliers based on univariate and multivariate criteria, the final dataset comprised 2269 participants. This sample size was deemed adequate for both CFA and invariance analyses. Detailed demographic information is provided in the Method section under Study Group 2.

**C-2 (10) Provide relevant statistical evidence about the construct equivalence, method equivalence, and item equivalence for all intended populations.**

To evaluate construct and method equivalence across subgroups, multi-group confirmatory factor analyses (MGCFA) were conducted based on three categorical variables: gender, personal computer ownership, and academic program. Gender was dichotomized as female/male, computer ownership as yes/no, and academic program was grouped into two-year and four-year programs.

The measurement invariance process followed a stepwise approach, examining configural, metric, and scalar models. Invariance was assessed using ΔCFI, ΔTLI, ΔRMSEA, and ΔSRMR, based on accepted thresholds [8,49]. Despite statistically significant changes in chi-square ($\chi^2$) between metric and scalar models, the changes in practical fit indices remained within acceptable limits, supporting the robustness and cross-group equivalence of the model.

Table 4 presents the discrepancies in fit indices and significance values (p) for the chi-square obtained in a measurement invariance analysis for three variables. The analysis was conducted on three variables: gender, possession of a personal computer, and program. The data were examined across two levels of invariance comparisons. The configural, metric, and scalar invariance models were tested. The indices comprise Chi-Square ($\chi^2$), p-values, Root Mean Square Error of Approximation (RMSEA), Comparative Fit Index (CFI), Tucker-Lewis Index (TLI), and Standardized Root Mean Square Residual (SRMR).

With regard to the variable of gender, the differences in the chi-square values are 9.625 (p = 0.217) for the configural-metric and 18.389 (p = 0.009) for the metric-scalar. The ΔRMSEA values demonstrate slight improvements, with differences of −0.003 and −0.034, respectively. The differences in the ΔCFI and ΔTLI are minimal, ranging from −0.001 to 0.004, indicating a stable fit across these levels. The ΔSRMR values remain largely unchanged, with differences of 0.001 and 0.000, respectively.

With regard to the possession of a personal computer, the differences in $\chi^2$ values are 10.605 (p = 0.168) for Configural-Metric and 27.256 (p = 0.003) for Metric-Scalar. The ΔRMSEA differences are −0.004 and −0.001, which indicate an improved fit at higher levels of invariance. The differences in the ΔCFI and ΔTLI values remain stable, ranging from −0.001 to 0.004. The ΔSRMR differences remain consistent at 0.001 for both comparisons.

Table 4. Summary of measurement invariance results for gender, computer ownership, and academic program.

| Variable | Fit Index | Configural Invariance | Metric Invariance | Scalar Invariance |
|---|---|---|---|---|
| Gender | Chi-Square | 654.972* | 664.597* | 682.986* |
| | df | 76 | 83 | 90 |
| | RMSEA | 0.082 (0.076 - 0.088) | 0.079(0.073 - 0.085) | 0.045 (0.039 - 0.050) |
| | CFI | 0.969 | 0.968 | 0.968 |
| | TLI | 0.954 | 0.958 | 0.961 |
| | SRMR | 0.05 | 0.051 | 0.051 |
| Possession of a personal computer | Chi-Square | 667.395* | 678.000* | 705.256* |
| | df | 76 | 83 | 90 |
| | RMSEA | 0.083 (0.077 - 0.089) | 0.079(0.073 - 0.085) | 0.078 (0.072 - 0.083) |
| | CFI | 0.968 | 0.967 | 0.966 |
| | TLI | 0.953 | 0.957 | 0.959 |
| | SRMR | 0.05 | 0.051 | 0.052 |
| Program | Chi-Square | 661.176* | 670.536* | 691.705 |
| | df | 76 | 83 | 90 |
| | RMSEA | 0.082 (0.076 - 0.088) | 0.079(0.073 - 0.085) | 0.077 (0.071 - 0.082) |
| | CFI | 0.968 | 0.968 | 0.967 |
| | TLI | 0.954 | 0.958 | 0.96 |
| | SRMR | 0.05 | 0.052 | 0.052 |

*p<0.05.

With regard to the Program variable, the differences in $\chi^2$ values are 9.360 (p=0.283) for Configural-Metric and 21.169 (p=0.004) for Metric-Scalar. The $\Delta$RMSEA differences are −0.003 and −0.002, indicating an enhancement in the model's fit. The differences in the $\Delta$CFI and $\Delta$TLI are minimal, ranging from −0.001 to 0.004. The $\Delta$SRMR differences are 0.002 and 0.000, indicating stable model fit across these levels.

These results indicate that while some significant changes in $\chi^2$ values occur, particularly for the Metric-Scalar comparison, the overall fit indices (RMSEA, CFI, TLI, and SRMR) show minimal changes, thereby supporting the robustness of the measurement model across different levels of invariance. These findings align with the criteria for assessing measurement invariance with small differences in fit indices [6].

**C-3 (11) Provide evidence supporting the norms, reliability, and validity of the adapted version of the test in the intended populations.**

The validity and reliability of the Turkish version of the scale were examined through a series of psychometric analyses using the data from Study Group 2 (N=2269). The analyses included confirmatory factor analysis (CFA), internal consistency estimation via Cronbach's alpha, discriminant validity assessment through HTMT2, and evaluation of item-total correlations. These procedures collectively aimed to ensure that the adapted scale retains construct and measurement integrity within the target culture.

**Construct validity – CFA results.** CFA was conducted to verify the four-factor structure of the adapted scale. The following model fit indices were obtained (see Table 5):

- RMSEA=0.079 (90% CI: 0.073–0.085), indicating intermediate fit

- CFI=0.964, TLI=0.947, and SRMR=0.049, all suggesting good fit

- $\chi^2$/df=15.150, exceeding conventional thresholds

**Table 5. Fit indexes of Transactional Distance Scale for target culture.**

| Fit Index | Range of Fit Indexes (Good&Intermediate fit) | Value | Interpretation |
|---|---|---|---|
| RMSEA | <0.08 | 0.079 (0.073 - 0.085) | Intermediate |
| CFI | >0.95 | 0.964 | Good |
| TLI | >0.95 | 0.947 | Good |
| SRMR | <0.06 | 0.049 | Good |
| Chi-Square/df | <2 | 15.150 | Poor |
| Chi-Square | | 575.673* | |
| df | | 38 | |

While the $\chi^2$/df ratio (15.150) substantially exceeds the traditional cutoff of 3, this statistic is known to be highly sensitive to large sample sizes, potentially exaggerating model misfit [28]. As recommended by Hu and Bentler [48], greater emphasis was placed on alternative fit indices, all of which suggest that the model fits the Turkish data well.

**Factor loadings and path diagram.** The standardized factor loadings ranged from.546 (Item y2) to.939 (Item y9), all statistically significant at p < .05. These values exceed the.50 threshold recommended for acceptable item–factor relationships [57], indicating strong loadings and satisfactory convergent validity. The explained variances ($R^2$) ranged from.122 to.702.

**The diagram presents the confirmatory factor analysis (CFA) model, showing factor–item loadings and inter-factor correlations. Items loaded as expected on their respective latent constructs: learner readiness (f1), effectiveness (f2), efficiency (f3), and satisfaction (f4).**

Fig 3 visually presents the CFA model structure. Factor-item relations are clearly delineated, and inter-factor correlations are depicted. Items loaded as expected on their respective latent dimensions: f1 (Learner Readiness), f2 (Effectiveness), f3 (Efficiency), and f4 (Satisfaction). Notably, Item y9 showed the highest loading (.939), while Item y2 had the lowest (.546), yet remained within acceptable bounds.

**Internal consistency and discriminant validity.** Internal reliability for each subdimension was assessed using Cronbach's alpha. As shown in Table 6, alpha values ranged from.702 to.939, exceeding the commonly accepted cutoff of.70 [56]. These results indicate that all subdimensions demonstrate satisfactory internal consistency.

Discriminant validity was evaluated via the Heterotrait-Monotrait Ratio of Correlations (HTMT2). All HTMT2 values between the latent dimensions were below the threshold of 0.90, providing evidence that each subscale measures distinct constructs [52,53].

The highest HTMT2 value was found between Learner Readiness and Satisfaction (0.879), but still remained under the.90 threshold, affirming discriminant validity across all pairs. To further assess item quality, item–total correlations were computed. The values ranged from.495 (Item 2) to.830 (Item 9), all of which exceeded the minimum recommended cutoff of.30 [39]. These results confirm that all items possess sufficient discrimination power, contributing reliably to their respective subscales.

In conclusion, the psychometric analyses provide robust support for the validity and reliability of the Turkish version of the Transactional Distance Scale. The model exhibited acceptable fit, strong factor loadings, internal consistency, discriminant validity, and item-level discrimination. These results indicate that the adapted version maintains its conceptual structure and measurement integrity across cultural boundaries.

**C-4 (12) Use an appropriate equating design and data analysis procedures when linking score scales from different language versions of a test.**

As the adaptation study did not include samples from multilingual populations or test versions in multiple languages, score linking procedures were not applied. The scale was administered in only one language to participants whose native

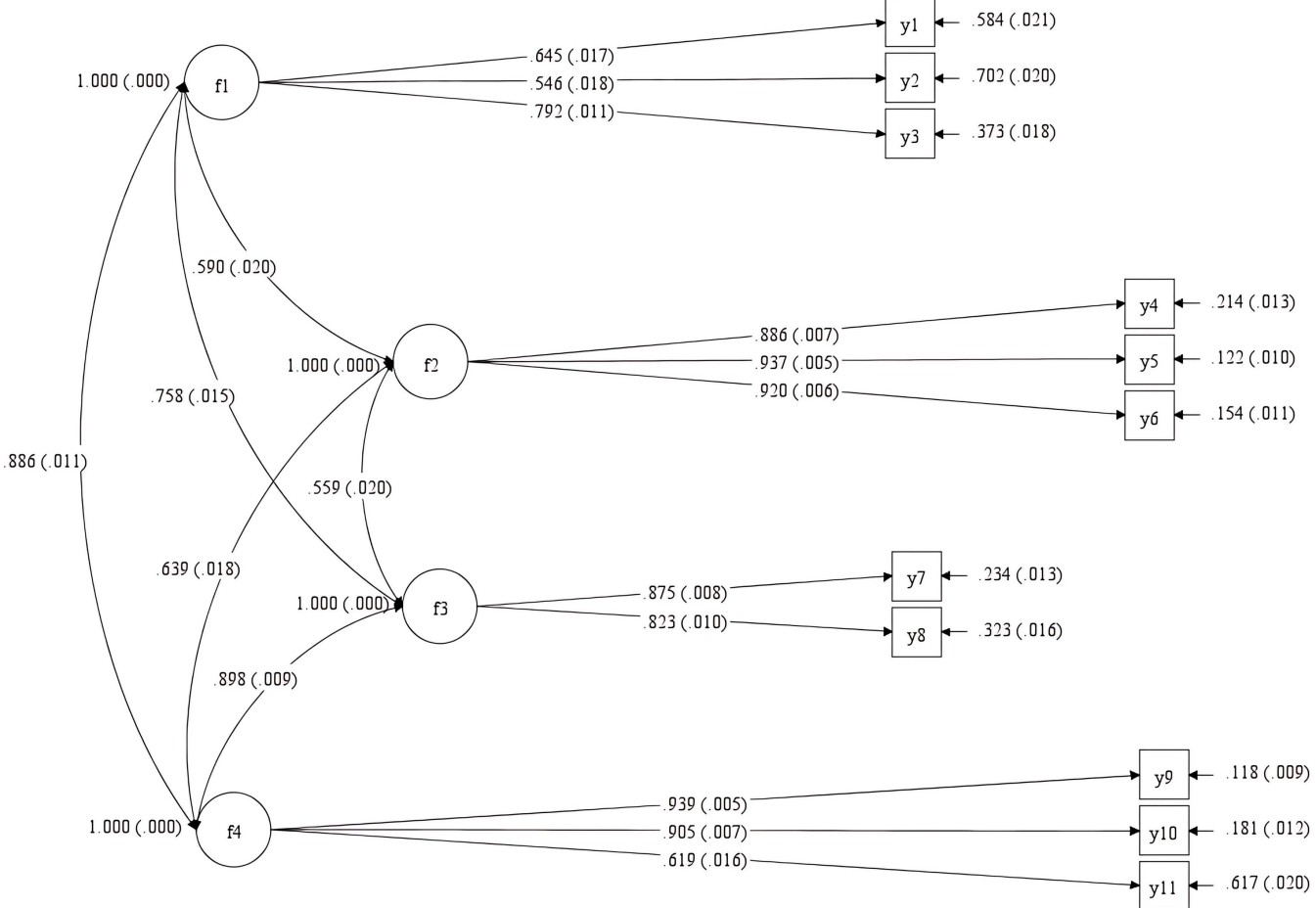

**Fig 3. Path diagram of the four-factor structure of the Turkish adaptation of the New TDSTECH scale.**

**Table 6. Cronbach Alpha and HTMT2 values for Transactional Distance Scale for Study Group 2.**

| Dimension | Cronbach Alpha | | HTMT2 | | | |
|---|---|---|---|---|---|---|
| | | | f1 | f2 | f3 | f4 |
| Learner Readiness For Using Technology (f1) | 0.702 | f1 | 1 | 0.601 | 0.746 | 0.879 |
| Effectiveness (f2) | 0.939 | f2 | – | 1 | 0.556 | 0.743 |
| Efficiency (f3) | 0.837 | f3 | – | – | 1 | 0.884 |
| Satisfaction (f4) | 0.852 | f4 | – | – | – | 1 |

language matched the target language. This step is acknowledged as a limitation and is proposed as a recommendation for future cross-linguistic adaptation studies.

**A-1 (13) Prepare administration materials and instructions to minimize any culture- and language-related problems that are caused by administration procedures and response modes that can affect the validity of the inferences drawn from the scores.**

To minimize potential cultural and linguistic misunderstandings during the implementation of the adapted scale, a guidance note was presented at the beginning of the questionnaire. This instructional material included:

- a concise definition of the transactional distance construct,

- detailed instructions on how to respond to each item,

- an explanation of the scoring format,

- an estimated completion time,

- ethical guidance regarding voluntary participation and data confidentiality, and

- contact information for the researchers responsible for the study.

This preface aimed to standardize the administration process and support the consistent application of the scale in future studies across similar populations..

**A-2 (14) Specify testing conditions that should be followed closely in all populations of interest**
The scale was found to be suitable for both online and paper-based administration. However, to ensure accessibility and accuracy of responses, researchers are advised to select the format based on participants' technological familiarity and digital literacy.

In online versions, it is not recommended to make all items mandatory, as this could lead to response bias or participant discomfort. Allowing participants to skip questions supports the ethical principle of autonomy and improves data authenticity.

During the initial application, the average response time ranged from 4 to 6 minutes, depending on participants' reading speed. Therefore, future administrations should allocate approximately 3 to 7 minutes for completion, accounting for individual variation.

**SSI-1 (15) Interpret any group score differences with reference to all relevant available information.**
To further support the linguistic equivalence of the scale, independent samples t-tests were conducted comparing the total scores obtained from the original and adapted forms administered to Study Group 1. The mean difference between the two versions was 0.522 points, which was not statistically significant ($t = 0.446$, $p > .05$).

This finding reinforces the conclusion that the source and target language versions of the scale are equivalent in terms of language and semantic meaning, thus satisfying the requirements of Step 15 of the ITC adaptation framework.

**SSI-2 (16) Only compare scores across populations when the level of invariance has been established on the scale on which scores are reported.**
Consistent with Step 12, this study did not include participants whose native language differed from the target language. Therefore, cross-population score comparisons were not feasible. It is recommended that future research address this limitation by including multilingual samples to enable such comparisons and further establish the scale's cross-cultural validity.

**Doc-1 (17) Provide technical documentation of any changes, including an account of the evidence obtained to support equivalence, when a test is adapted for use in another population.**
A comprehensive technical report was prepared to document all modifications and findings throughout the adaptation process. This report includes detailed tables summarizing validity and reliability analyses, aligned with the guidelines of ITC adaptation standards. The technical report has been appended to the study as Appendix-1, serving as a resource for researchers and practitioners interested in the methodological rigor of the adaptation.

**Doc-2 (18) Provide documentation for test users that will support good practice in the use of an adapted test with people in the context of the new population.**
To facilitate proper and standardized use of the adapted scale in diverse settings, a concise application guide was developed. This guide outlines the administration procedures, scoring instructions, and practical recommendations for future users. It aims to promote consistency and fidelity in the scale's deployment among researchers, educators, and practitioners. The guide is included as Appendix-2 at the end of the study.

## Discussion and conclusion

This study provides a rigorous adaptation of the New TDSTECH scale to the Turkish educational context, assessing its psychometric properties among university students. The results affirm the scale's four-factor structure, demonstrating strong internal consistency, discriminant validity, and construct validity. These findings are consistent with prior research emphasizing the central role of student-technology interaction in shaping transactional distance within online learning environments [9,16]. Moreover, this adaptation enriches the measurement tools available for Turkish educators and researchers to assess and enhance learner engagement with diverse educational technologies.

Prior Turkish studies such as Horzum [23] have contributed valuable insights into the measurement of transactional distance, developing scales with multiple subdimensions including dialogue, autonomy, structure flexibility, content organization, and student control. However, these earlier instruments do not capture the evolving technological interface interaction dimension now central to understanding distance learning experiences. This highlights the necessity and value of the current adaptation, which integrates this critical contemporary component.

### Theoretical framework and contemporary relevance

The concept of transactional distance has evolved as a pivotal theoretical lens in understanding distance education, encompassing the cognitive, affective, and technological dimensions that influence learner experiences [10]. Recent advancements in educational technology—particularly the proliferation of learning management systems, synchronous and asynchronous platforms, and mobile learning—have necessitated refined instruments that capture the nuanced student-interface interactions [16,18]. The New TDSTECH scale's incorporation of this dimension responds directly to this need, reflecting a comprehensive understanding of transactional distance as it manifests in contemporary, technology-mediated learning environments.

The adaptation to Turkish culture underscores the universal relevance of the scale's constructs while acknowledging contextual particularities. Turkey's rapid digital transformation in education, accelerated by the COVID-19 pandemic, situates this instrument as timely and essential for evaluating the shifting dynamics of online learning 3]. The scale's ability to measure learner readiness, perceived effectiveness, efficiency, and satisfaction offers stakeholders actionable insights into the interplay between students and technological interfaces.

### Cultural context and adaptation challenges

Despite the scale's robust psychometric properties, cultural nuances inherent to the Turkish educational landscape pose considerations for interpretation and application. Regional disparities in digital access and literacy reflect broader socio-economic inequalities, which may influence transactional distance perceptions beyond the scope of the scale [22]. The adaptation process addressed linguistic equivalences and cultural references meticulously, as exemplified by the inclusion of diverse platform names such as LMS, Google Classroom, and MS Teams to reflect localized technological environments.

Furthermore, educational norms, pedagogical practices, and student expectations vary across cultural contexts, potentially affecting how transactional distance components are experienced and reported [41]. While this study's sample consisted primarily of university students with moderate to high digital proficiency, extending research to include rural populations, younger learners, or adult education contexts could uncover differentiated patterns of interaction and distance.

### Practical implications and policy recommendations

The validated Turkish adaptation of the New TDSTECH scale holds significant promise for informing educational policy and practice. Educators can utilize the instrument to diagnose areas where learner-technology interactions may hinder engagement, allowing targeted interventions such as digital literacy programs or platform usability improvements.

Institutions may also apply the scale to monitor the impact of technological innovations or pedagogical reforms on student experiences.

At the policy level, findings highlight the necessity of equitable access to technology and support structures to minimize transactional distance disparities, particularly for underserved populations. Incorporating regular assessments using culturally validated instruments can facilitate evidence-based decisions aimed at enhancing distance education quality nationwide.

### Methodological considerations and future research directions

While this study's methodological rigor, including large sample size and adherence to ITC adaptation standards, strengthens confidence in the findings, certain limitations remain. The reliance on a university student sample limits generalizability to other educational levels and demographic groups. Additionally, the absence of multilingual samples precluded cross-cultural measurement invariance analyses beyond gender, program, and computer ownership.

While the $\chi^2$/df ratio (15.150) substantially exceeds the conventional threshold (< 3), this statistic is known to be highly sensitive to large sample sizes, potentially exaggerating model misfit [28]. Complementary fit indices such as CFI, TLI, RMSEA, and SRMR provide a more balanced and reliable assessment of model adequacy, supporting the structural validity of the scale in the Turkish context. Therefore, caution should be exercised in interpreting this index in isolation.

Future research should prioritize longitudinal studies to examine how transactional distance perceptions evolve over time and in response to changing technologies. Expanding the scope to include K-12 students, adult learners, and individuals with varying digital competencies will enhance the scale's applicability. Cross-cultural comparative studies involving multiple language adaptations can further elucidate the scale's universality and sensitivity to cultural nuances.

Moreover, qualitative investigations complementing quantitative findings may deepen understanding of the contextual factors influencing transactional distance, such as institutional support, learner motivation, and socio-cultural attitudes toward technology.

## Conclusion

This study successfully demonstrates that the Turkish adaptation of the New TDSTECH scale is a psychometrically sound instrument capable of reliably measuring transactional distance between students and technology in online learning settings. Its validated four-factor structure captures critical dimensions influencing learner engagement and satisfaction, offering valuable tools for educators, researchers, and policymakers.

As digital education continues to evolve rapidly, especially in response to global disruptions such as the COVID-19 pandemic, instruments like the New TDSTECH are essential for guiding improvements in educational design and delivery. By facilitating nuanced assessments of learner-technology interactions, this scale supports efforts to create more effective, equitable, and engaging online learning experiences.

Acknowledging the study's limitations, ongoing research must aim to broaden the scale's applicability and refine its sensitivity to diverse learner populations and technological contexts. Such endeavors will contribute substantially to the advancement of distance education theory and practice in Turkey and beyond.

## Supporting information

**S1 Appendix. Technical report providing detailed documentation of the adaptation steps, statistical analyses, and psychometric validation results in accordance with ITC guidelines.**
(DOCX)

**S2 Appendix. Application guide outlining administration procedures, scoring instructions, and recommendations for using the Turkish version of the New TDSTECH scale in educational contexts.**
(DOCX)

## Author contributions

**Conceptualization:** Alperen YANDI.

**Data curation:** Alperen YANDI.

**Formal analysis:** Alperen YANDI.

**Funding acquisition:** Alperen YANDI.

**Investigation:** Alperen YANDI.

**Methodology:** Alperen YANDI.

**Project administration:** Alperen YANDI.

**Resources:** Alperen YANDI.

**Software:** Alperen YANDI.

**Supervision:** Alperen YANDI.

**Validation:** Alperen YANDI.

**Visualization:** Alperen YANDI.

**Writing – original draft:** Alperen YANDI.

**Writing – review & editing:** Alperen YANDI.

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
