## [Decision Letter · Decision Letter 0]

3 Aug 2025

Dear Dr. YANDI,

Thank you for submitting your manuscript to PLOS ONE. After careful consideration, we feel that it has merit but does not fully meet PLOS ONE’s publication criteria as it currently stands. Therefore, we invite you to submit a revised version of the manuscript that addresses the points raised during the review process.

We look forward to receiving your revised manuscript.

Kind regards,

Zülbiye KAÇAY, assoc. prof.

Academic Editor

PLOS ONE

2. Please note that your Data Availability Statement is currently missing [the repository name and/or the DOI/accession number of each dataset OR a direct link to access each database]. If your manuscript is accepted for publication, you will be asked to provide these details on a very short timeline. We therefore suggest that you provide this information now, though we will not hold up the peer review process if you are unable.

Additional Editor Comments:

Thank you for submitting your manuscript, “Turkish Adaptation of The New TDSTECH Scale”, to PLOS ONE. Your work addresses an important and timely topic, and your commitment to methodological rigor, particularly the adherence to ITC’s test adaptation guidelines, is commendable.

Both reviewers acknowledged the technical strength of your study, as well as the relevance and originality of your contribution. However, to improve the clarity, readability, and completeness of your manuscript, several substantial revisions are necessary before it can be considered for publication.

Please consider the following essential points in your revision:

Language and Clarity: The manuscript requires comprehensive language editing by a native English speaker or professional editing service. This includes improving sentence structure, eliminating redundancy, and enhancing overall clarity.

Abstract:

Avoid repetitive sentences.

Include key statistical results from your analyses.

Do not cite or mention the original scale in the abstract.

Title: Avoid using acronyms (e.g., TDSTECH) in the title. Use the full name of the scale instead.

Formatting:

Remove the Biography section (not part of PLOS ONE article structure).

Use reference citation numbers in square brackets, per journal style (e.g., [12]).

Introduction: Shorten the section for better readability. Aim for clearer and more concise paragraph structure.

Method Section:

Move the data collection period from the Ethics section to the beginning of the Methods.

Streamline overly detailed descriptions; some procedural information is better summarized.

Clearly explain how the sample size was determined.

Ensure that all analysis results (e.g., CFA fit indices) are moved to the Results section.

Results Section:

Address the high chi-square/df ratio (15.15), which diverges from fit standards, even though other indices are acceptable. This discrepancy should be acknowledged and discussed.

Discussion Section:

Currently missing. Please include a well-developed discussion that compares your findings with existing literature, explains your model fit outcomes, and provides practical implications.

Conclusion:

Go beyond summarizing the study. Include insights on the significance of findings and recommendations for future research.

Student-Interface Interaction:

Clarify and contextualize this concept specifically within Turkish distance education culture, as suggested by Reviewer 2.

Technical Edits:

Page 29 appears blank—please review and correct.

We appreciate your contribution and look forward to receiving a carefully revised version of your manuscript that addresses these issues in full.

Reviewers' comments:

Reviewer's Responses to Questions

**Comments to the Author**

1. Is the manuscript technically sound, and do the data support the conclusions?

Reviewer #1: Yes

Reviewer #2: Yes

2. Has the statistical analysis been performed appropriately and rigorously?

Reviewer #1: Yes

Reviewer #2: Yes

3. Have the authors made all data underlying the findings in their manuscript fully available?

Reviewer #1: No

Reviewer #2: Yes

4. Is the manuscript presented in an intelligible fashion and written in standard English?

Reviewer #1: No

Reviewer #2: Yes

Reviewer #1: Thank you for your submission. Your study addresses a timely and relevant topic—the adaptation of the new TDSTECH scale to Turkish culture in the context of distance education.

The research is grounded in a solid theoretical framework, follows internationally recognized test adaptation guidelines, and applies robust statistical analyses with a large and diverse sample. These are significant strengths that enhance the study's credibility.

However, the manuscript requires major revisions before it can be considered for publication. Below are my detailed comments and suggestions:

-It is recommended that the article should be reviewed by a native English speaker.

-It may be more appropriate to avoid repetitive sentences in the abstract and instead provide numerical data related to the results.

-It is not necessary to cite or include the original article on the scale in the abstract.

-According to the journal submission guideline, biography is not one of the sections of the article.

-The introduction is too long. The paragraphs themselves are too long. Short, clear, and simple sentences will increase the readability of the text.

-There should be no acronyms in the title.

- In the text, cite the reference number in square brackets (e.g., “We used the techniques developed by our colleagues [19] to analyze the data”).

-A short, clear sentence will suffice for the aim of the article.

-The method section is quite detailed. However, the first paragraph of this section practically introduces the method. Then, a brief summary of the guide to the method is provided. Instead, simply describing the steps you followed based on the guide you referenced would suffice. Otherwise, reading the text becomes lengthy and tedious.

-Some analyses are overly detailed in the method section.

-The data collection period for the study should be stated at the beginning of the method section, not in the ethics statement.

-Despite a very detailed method section, it is not specified how the sample size is determined.

-Analysis results that cannot belong to the method such as CFA should be given in the results section.

-Does Scientiæ Baccalaureus mean students studying in a 4-year degree program? Please use a more up-to-date equivalent (Bachelor's degree students).

-Like the rest of the article, the results section is systematic and methodologically successful, but at times too detailed and technical.

-Page 29 is just a blank page.

-While the CFA results overall suggest an acceptable model fit (e.g., CFI = 0.964; TLI = 0.947; RMSEA = 0.079; SRMR = 0.049), the reported chi-square/df ratio (15.150) is substantially above the conventional thresholds (typically < 3 for acceptable fit). I would recommend the authors address this inconsistency more explicitly in the results or discussion section.

-There is no discussion part in the article. In the discussion section, topics such as comparison of the results with the literature, limitations of the study, comparison with previous scale studies should be included.

-The conclusion is more of a summary of the study. In this section, inferences such as the importance of the findings and their impact should be made instead of a summary.

-The limitations of the study should be developed and included in the discussion in this section and suggestions for future studies based on these should be addressed in the conclusion section.

I wish you success in your work.

Reviewer #2: Some terms, like "student-interface interaction," are mentioned but not explained in detail. It would be helpful to provide more context or examples to clarify what this term means and why it’s significant in the context of Turkish culture.

Suggestion: Include a brief explanation of "student-interface interaction" and its relevance to the study, especially in the Turkish educational context.

**Do you want your identity to be public for this peer review?** For information about this choice, including consent withdrawal, please see our Privacy Policy

Reviewer #1: No

Reviewer #2: No

---

## [Author Response · Author response to Decision Letter 1]

19 Aug 2025

Point-by-Point Response to Reviewers and Editors

Reviewer / Editor Comment: PLOS ONE style compliance: Ensure manuscript follows journal style, including file naming, reference formatting, and section order.

Author Response: Revised the manuscript fully in accordance with PLOS ONE style templates. Corrected section headings, table and figure captions, reference style (numbers in square brackets), and removed any sections not in journal structure. Updated file names accordingly.

Reviewer / Editor Comment: Data Availability Statement incomplete: Repository name and DOI missing.

Author Response: All datasets, R and Mplus syntax files, and supplementary documents have been deposited in Zenodo. Updated Data Availability Statement to read: “All relevant data are available in the Zenodo repository: Yandı, A. (2025). Turkish Adaptation of a New Scale for Measuring Transactional Distance Between Students and the Learning Technology [Data set]. Zenodo. https://doi.org/10.5281/zenodo.16880975.”

Reviewer / Editor Comment: Data sharing plan clarification: Entire dataset must be accessible upon acceptance.

Author Response: Confirmed full open access to anonymized datasets and analysis scripts in Zenodo. Statement revised to reflect compliance with PLOS ONE open data policy.

Reviewer / Editor Comment: Ethics statement placement: Must only appear in Methods section.

Author Response: Ethics statement relocated to the Methods section exclusively. Removed from all other parts of the manuscript.

Reviewer / Editor Comment: Language and clarity: Comprehensive editing by a native/professional editor required.

Author Response: The entire manuscript underwent professional English language editing to improve grammar, sentence structure, clarity, and remove redundancies.

Reviewer / Editor Comment: Abstract – avoid repetition, include key statistical results, remove citation of original scale.

Author Response: Abstract rewritten to remove repetitive content, add key statistical indicators (e.g., CFI, TLI, RMSEA, SRMR, Cronbach’s α), and exclude any mention of the original scale.

Reviewer / Editor Comment: Title – avoid acronyms such as TDSTECH.

Author Response: Title revised to spell out the full name: “Turkish Adaptation of a New Scale for Measuring Transactional Distance Between Students and the Learning Technology.”

Reviewer / Editor Comment: Remove Biography section (not part of PLOS ONE structure).

Author Response: Biography section completely removed.

Reviewer / Editor Comment: Introduction too long – improve paragraph structure and conciseness.

Author Response: Introduction shortened by removing redundant sentences, reorganized into shorter, coherent paragraphs with smoother transitions.

Reviewer / Editor Comment: Aim statement should be short and clear.

Author Response: Rephrased the aim into a single concise sentence at the end of the Introduction.

Reviewer / Editor Comment: Method section – move data collection period to beginning.

Author Response: Added the data collection period (Nov 29–Dec 25, 2021) at the start of the Method section.

Reviewer / Editor Comment: Method section – streamline overly detailed descriptions.

Author Response: Condensed lengthy procedural descriptions while keeping essential methodological information; reduced tedious procedural repetition.

Reviewer / Editor Comment: Clearly explain sample size determination.

Author Response: Added literature-based justification for sample size, citing MacCallum et al. (1999), Wolf et al. (2013), and Kline (2018).

Reviewer / Editor Comment: Move CFA results from Method to Results section.

Author Response: Relocated all CFA model fit results, factor loadings, and related analysis details to the Results section.

Reviewer / Editor Comment: Clarify “Scientiæ Baccalaureus” term.

Author Response: Updated term to “Bachelor’s degree students.”

Reviewer / Editor Comment: Blank page (p. 29) in manuscript.

Author Response: Deleted the blank page.

Reviewer / Editor Comment: CFA χ²/df ratio high – discuss.

Author Response: Discussion now explains sensitivity of χ²/df to large sample sizes and emphasizes evaluation using alternative fit indices (CFI, TLI, RMSEA, SRMR).

Reviewer / Editor Comment: No Discussion section – include comparisons, implications, and limitations.

Author Response: Added a full Discussion section comparing findings to prior literature, interpreting model fit, and outlining both practical implications and study limitations.

Reviewer / Editor Comment: Conclusion too brief – extend beyond summary.

Author Response: Conclusion expanded to highlight significance of findings, implications for practice, and recommendations for future research.

Reviewer / Editor Comment: Limitations and future research suggestions should be developed.

Author Response: Expanded limitations to include lack of multilingual samples, university-only sample, and contextual constraints; added explicit recommendations for future work.

Reviewer / Editor Comment: Clarify “student-interface interaction” in Turkish distance education context.

Author Response: Added explanation in Introduction and Discussion contextualizing this concept in Turkey’s distance education culture with relevant examples.

Reviewer / Editor Comment: Ensure references follow PLOS ONE style and citation numbering.

Author Response: Revised all in-text citations to numerical format; reformatted reference list per PLOS ONE guidelines, verifying and adding DOIs where available.

Reviewer / Editor Comment: Check relevance of suggested citations.

Author Response: Reviewed all suggested works; included only those directly relevant to the study’s aims and context.

Reviewer / Editor Comment: Technical checks (figures/tables).

Author Response: Reformatted all tables and figures to comply with journal layout, numbering, and caption standards.

Reviewer #1 Comment: Manuscript technically sound but requires major revisions for readability and completeness.

Author Response: Addressed by implementing all specific revisions above to improve clarity, reduce redundancy, and ensure completeness of all sections.

Reviewer #1 Comment: Overly detailed Method section with lengthy step-by-step guide from ITC adaptation.

Author Response: Condensed ITC steps where possible without losing methodological rigor; preserved critical details while reducing text volume.

Reviewer #1 Comment: Some analyses overly detailed in Method section.

Author Response: Moved analytical results out of Method and into Results; summarized procedures in Method.

Reviewer #1 Comment: Replace “Moodle” with contextually appropriate platforms.

Author Response: Replaced “Moodle” with “online learning platforms (e.g., LMS, Moodle, MS Teams, Google Classroom)” in relevant items.

Reviewer #1 Comment: Clarify limitations and their implications.

Author Response: Limitations expanded in Discussion, with corresponding implications integrated into Conclusion.

Reviewer #2 Comment: Explain ‘student-interface interaction’ term in detail.

Author Response: Provided clear definition and examples tailored to Turkish online education context, illustrating its significance for learner engagement.

---

## [Editor Report · Decision Letter 1]

21 Aug 2025

Turkish Adaptation of a New Scale for Measuring Transactional Distance Between Students and the Learning Technology

PONE-D-25-35376R1

Dear Dr. Alperen YANDI,

We’re pleased to inform you that your manuscript has been judged scientifically suitable for publication and will be formally accepted for publication once it meets all outstanding technical requirements.

Kind regards,

Zülbiye KAÇAY, assoc. prof.

Academic Editor

PLOS ONE

Additional Editor Comments (optional):

Thank you for your careful and thorough revision of the manuscript. You have addressed all reviewer and editorial concerns point by point, and the revised version shows significant improvement in structure, clarity, and compliance with PLOS ONE guidelines. The addition of a comprehensive Discussion section, expanded Conclusion, and clear statement of limitations have strengthened the manuscript.

The updated Data Availability Statement with an open-access Zenodo DOI, along with professional English editing and improved formatting of tables, figures, and references, ensures that the article is ready for publication.

I am pleased to recommend your manuscript for acceptance in PLOS ONE. Congratulations on this achievement.
---

## [Editor Report · Acceptance letter]

PONE-D-25-35376R1

PLOS ONE

Dear Dr. YANDI,

I'm pleased to inform you that your manuscript has been deemed suitable for publication in PLOS ONE. Congratulations! Your manuscript is now being handed over to our production team.

Kind regards,

on behalf of

Professor Zülbiye KAÇAY

Academic Editor

PLOS ONE